# Images that Sound:
# Composing Images and Sounds on a Single Canvas

**Ziyang Chen**    **Daniel Geng**    **Andrew Owens**

University of Michigan

https://ificl.github.io/images-that-sound/

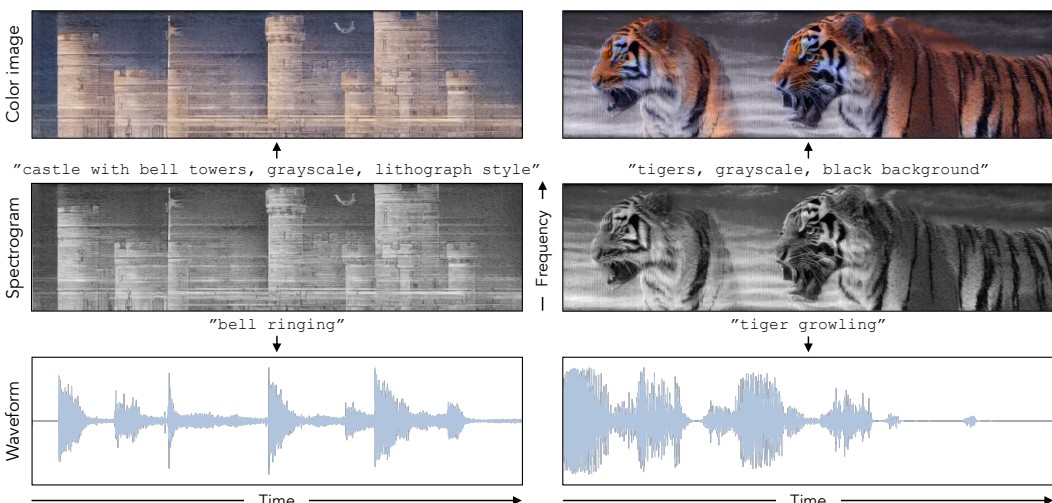

Figure 1: **Images that sound.** We use diffusion models to generate visual spectrograms (second row) that look like natural images, which we call *images that sound*. These spectrograms can be converted into natural sounds (third row) using a pretrained vocoder, or colorized to obtain more visually pleasing results (first row). Please refer to our website to listen to the sounds.

## Abstract

Spectrograms are 2D representations of sound that look very different from the images found in our visual world. And natural images, when played as spectrograms, make unnatural sounds. In this paper, we show that it is possible to synthesize spectrograms that simultaneously look like natural images and sound like natural audio. We call these visual spectrograms *images that sound*. Our approach is simple and zero-shot, and it leverages pre-trained text-to-image and text-to-spectrogram diffusion models that operate in a shared latent space. During the reverse process, we denoise noisy latents with both the audio and image diffusion models in parallel, resulting in a sample that is likely under both models. Through quantitative evaluations and perceptual studies, we find that our method successfully generates spectrograms that align with a desired audio prompt while also taking the visual appearance of a desired image prompt. Please see our project page for video results: https://ificl.github.io/images-that-sound/

## 1 Introduction

The spectrogram is a ubiquitous low-dimensional representation for audio machine learning that plots the energy within different frequencies over time. But it is also widely used as a tool for converting

38th Conference on Neural Information Processing Systems (NeurIPS 2024).

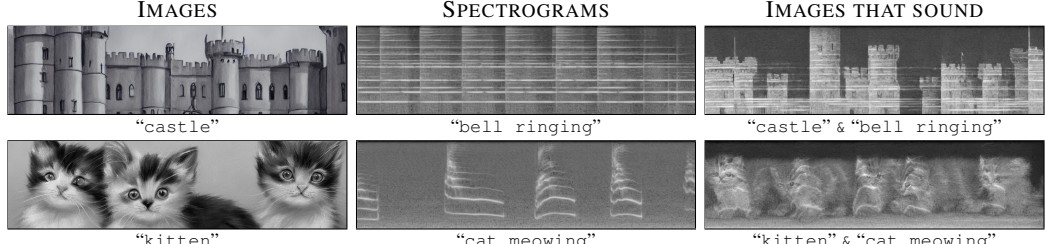

| IMAGES | SPECTROGRAMS | IMAGES THAT SOUND |
|---|---|---|
| "castle" | "bell ringing" | "castle" & "bell ringing" |
| "kitten" | "cat meowing" | "kitten" & "cat meowing" |

Figure 2: **Images vs. spectrograms.** We show grayscale images generated from Stable Diffusion [96] on the left, followed by log-mel spectrograms generated from Auffusion [118] in the middle, and our generated *images that sound* results on the right.

sound into a visual form that can be—at least partially—perceived by sight. For example, in this representation (Fig. 2), event onsets look to a human observer like lines, and speech looks like a sequence of waves and bands. This insight is commonly used within the audio community, which frequently repurposes pretrained visual networks for audio tasks, often with only relatively minor modifications [48, 90, 69, 68, 34, 118, 112].

We hypothesize that the success of spectrograms in these roles is due in part to the fact that they share many statistical properties with the distribution of natural images, providing visual structures like edges and textures that the human visual system can readily process. Given the statistical similarities between images and sounds, we ask whether it is possible to automatically generate examples that lie at the *intersection* of both modalities. We create *images that sound* (Fig. 1), 2D matrices that *look* semantically meaningful when viewed as images, but that also *sound* meaningful when played as a spectrogram. This generative modeling problem is challenging, because it requires modeling a distribution that is induced by two very different data sources, and no relevant paired data is available.

We are motivated by the "spectrogram art" that has been made by a variety of artists [10], most famously by musicians Aphex Twin [2], Venetian Snares [113] and Nine Inch Nails [85]. These artists manipulate their songs to display a desired image when they are visualized as spectrograms, such as by showing the artist's face or album art. In current practice, there is a steep trade-off between the quality of the image and the sound, since it is difficult to simultaneously control the interpretation of a signal in both modalities. As a result, existing artwork often comes across to the listener as dissonant or as random noise, rather than as natural sounds.[1] By contrast, we aim to generate signals that are as natural as possible in both modalities, such as towers that simultaneously sound like ringing bells or images of tigers that make a roaring sound (Fig. 1).

In this work, we pose this problem as a multimodal compositional generation task and propose a simple, zero-shot method that composes off-the-shelf text-to-spectrogram and text-to-image diffusion models from different modalities. Inspired by prior work on compositionality in diffusion models [72, 29, 42, 41], we denoise using both a noise estimate from the spectrogram model and a noise estimate from the image model. This is possible because these two models perform diffusion in the same latent space. The result is a sample that is simultaneously likely under the (text-conditional) distribution of spectrograms and images. The spectrograms are then converted to waveforms using a pretrained vocoder. In addition, we show that these black-and-white images may be colorized, resulting in color images whose grayscale versions can be played as spectrograms.

Surprisingly, we find that off-the-shelf diffusion models *trained on different modalities* can be composed together to obtain samples that function as both an image and a sound. Often these examples reuse visual elements in unexpected ways (*e.g.*, in Fig. 1, a line is both the onset of a bell chime and the contour of a bell tower). We provide qualitative results, as well as quantitative comparisons and human study results against baselines, indicating that our method produces spectrograms that better align with both the audio and image prompts. Our contributions are summarized as follows:

- We propose *images that sound*, a type of multimodal art that can be both understood as an image or played as a sound.
- We show that we can compose pretrained diffusion models from different modalities in a zero-shot fashion to produce examples at the intersection of image and spectrogram distributions.

---

[1]We encourage the reader to listen to popular examples of spectrogram art [10].

- We propose alternative methods for generating *images that sound*, one based on score distillation sampling [11, 93] and another based on simply subtracting an image from a spectrogram.
- We find through qualitative and quantitative experiments that our method outperforms baseline approaches and generates high-quality samples.

## 2   Related Work

**Diffusion models.**   Diffusion models [102, 54, 106, 27, 104] are a class of generative models that learn to reverse a forward process that iteratively corrupts data. Typically, this forward process adds Gaussian noise and the reverse process learns to denoise the data by predicting the added noise. Diffusion models have a variety of applications, including text-conditioned image generation [27, 96, 84, 26, 98], video generation [53, 56, 101, 6, 45, 115], image and video editing [94, 97, 79, 49, 9, 31, 40], audio generation [118, 70, 71, 32, 43, 76], 3D generation [74, 57, 12, 73, 8, 38], and camera pose estimation [121]. In this work, we use Stable Diffusion [96], a latent diffusion model trained for text-conditioned image generation, as well as Auffusion [118], a text-conditioned audio generation model trained to produce log-mel spectrograms. Auffusion is finetuned from Stable Diffusion, similar to Riffusion [34], and as a result, the two methods share a latent space. This is crucial for our technique, which jointly diffuses these shared latents.

**Compositional generation.**   One property of diffusion models is that they admit a straightforward technique to compose concepts by summing noise estimates. This may be understood by viewing noise estimates as gradients of a conditional data distribution [105, 106], and the sum of these gradients as pointing in the direction that maximizes multiple conditional likelihoods. This approach has been applied to enable compositions of text prompts globally [72], spatially [7, 29], transformations of images [42], and image components [41]. We go beyond these works by showing that diffusion models *from two different modalities* can successfully be composed together.

**Audio-visual learning.**   A variety of works have learned cross-modal associations between vision and sound. Some approaches establish *semantic correspondence*, *i.e.*, which sounds and visuals are commonly associated with one another [3, 108]. Previous work has used this cue to learn cross-modal representations [5, 83, 80, 46, 44, 69, 68] and audio-visual sound localization [4, 58, 81, 59, 100, 91, 75]. Some researchers focus on the *temporal correspondence* between audio and visual streams [64, 87, 33, 16, 107, 62] to study source separation [122, 1, 37], Foley sound generation [61, 28, 117, 78], and action recognition [39, 60, 86]. Others also explore the *spatial correspondence* between them [21, 35, 120, 19, 22, 77], including spatial sound generation [36, 82, 14, 67] and audio-visual acoustic learning [13, 103, 24, 18, 20]. Differing from the works above, our focus is to explore the intersection of the distributions between spectrograms and images, where we create spectrograms that can be understood as visual images and can also be played as sounds.

**Audio steganography.**   Audio steganography is the practice of concealing information within an audio signal. Artists have explored it for creative expression [111, 110]. Aphex Twin embedded a visual of his face in the audio waveform of the track "Formula" [2]. Noam Oxman creates animal portraits made of musical notations [88]. Other work has proposed deep learning methods for steganography, such as hiding video content inside audio files with invertible generative models [119], hiding audio data inside an identity image [123], and audio watermarking [15, 89, 99]. Our approach can be viewed as a steganography method that hides an image within an audio track, and is only revealed when the track is converted to a spectrogram.

## 3   Method

Our goal is to generate spectrograms that simultaneously represent both a sound and an image, each of which is specified by a text prompt. When the spectrogram is converted into a waveform, the sound matches the audio prompt, while when it is visually inspected, it should take the appearance of a given visual prompt (Fig. 1). To do this, we sample from the joint distribution of images and spectrograms, using off-the-shelf diffusion models trained on each modality independently.

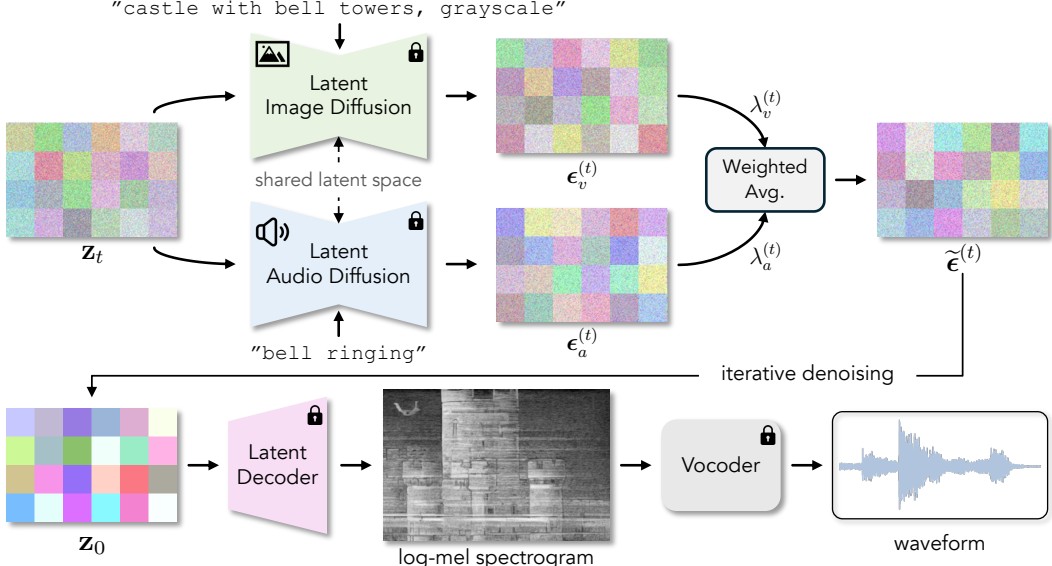

Figure 3: **Composing audio and visual diffusion models.** We generate the visual spectrogram that can be visualized as an image or played as a sound. Given a noisy latent $\mathbf{z}_t$, we apply visual and audio diffusion models, each guided by a text prompt, to compute noise estimates $\boldsymbol{\epsilon}_v^{(t)}$ and $\boldsymbol{\epsilon}_a^{(t)}$ respectively. We obtain the multimodal noise estimate $\tilde{\boldsymbol{\epsilon}}^{(t)}$ by a weighted average, then use it as part of the iterative denoising process. Finally, we decode the clean latent $\mathbf{z}_0$ to a spectrogram and convert it into a waveform using a pretrained vocoder (or by Griffin-Lim [47]).

## 3.1 Preliminaries

**Diffusion models.** Diffusion models [54, 106] iteratively denoise standard Gaussian noise, $\mathbf{x}_T \sim \mathcal{N}(\mathbf{0}, \mathbf{I})$, to generate clean samples, $\mathbf{x}_0$, from some learned data distribution. At timestep $t$ in the reverse diffusion process, the noise predictor, $\boldsymbol{\epsilon}_\theta$, takes the intermediate noisy sample, $\mathbf{x}_t$, and the condition $y$, such as a text prompt embedding, to estimate the noise $\boldsymbol{\epsilon}_\theta(\mathbf{x}_t; y, t)$. Following DDIM [104], we obtain the next, less noisy, sample $\mathbf{x}_{t-1}$ at the previous timestep via:

$$\mathbf{x}_{t-1} = \sqrt{\alpha_{t-1}} \left( \frac{\mathbf{x}_t - \sqrt{1 - \alpha_t} \cdot \hat{\boldsymbol{\epsilon}}_\theta(\mathbf{x}_t; y, t)}{\sqrt{\alpha_t}} \right) + \sqrt{1 - \alpha_{t-1} - \sigma_t^2} \cdot \hat{\boldsymbol{\epsilon}}_\theta(\mathbf{x}_t; y, t) + \sigma_t \boldsymbol{\epsilon}_t, \quad (1)$$

where $\boldsymbol{\epsilon}_t$ is independent Gaussian noise, $\alpha_t$ is a predefined coefficient, and $\sigma_t$ controls the randomness level which we set to 0 for deterministic sampling. We may also optionally apply classifier-free guidance (CFG) [55] by modifying the noise estimate as:

$$\hat{\boldsymbol{\epsilon}}_\theta(\mathbf{x}_t; y, t) = \boldsymbol{\epsilon}_\theta(\mathbf{x}_t; \varnothing, t) + \gamma \left( \boldsymbol{\epsilon}_\theta(\mathbf{x}_t; y, t) - \boldsymbol{\epsilon}_\theta(\mathbf{x}_t; \varnothing, t) \right), \quad (2)$$

where $\gamma$ denotes the strength of the conditional guidance and $\varnothing$ is the unconditional embedding of the empty string. This often results in much higher-quality samples.

**Latent diffusion.** Latent Diffusion Models (LDMs) [96] perform the diffusion process in a latent space rather than in pixel space. A pretrained encoder and decoder pair, $\mathcal{E}$ and $\mathcal{D}$, translates between pixel space and latent space. The latent space is typically much more compact and information-dense, which makes diffusion in this space more efficient. We use pretrained LDMs in our approach, due to the availability of audio and visual models with the same latent space.

## 3.2 Multimodal Denoising

Our goal is to generate an example $\mathbf{x} \in \mathbb{R}^{m \times n}$ that would be likely to appear under both visual and audio distributions, $p_a(\cdot)$ and $p_v(\cdot)$. We formulate this as sampling from a product of expert models[2] [52]: $p_{av}(\mathbf{x}) \propto p_a(\mathbf{x}) p_v(\mathbf{x})$. We follow recent work on the compositional generation that

---

[2]Recent work has called this a *conjunction* [30], since conceptually the samples are roughly from the intersection of both distributions.

samples from this distribution using the score functions from pretrained diffusion models [30]. In contrast to these approaches, however, our two models are trained on *two different modalities*.

We create our spectrograms using two pretrained latent diffusion models. One trained to generate images, $\boldsymbol{\epsilon}_{\phi,v}(\cdot,\cdot,\cdot)$, and the other to generate spectrograms, $\boldsymbol{\epsilon}_{\phi,a}(\cdot,\cdot,\cdot)$, both operating in the same latent space. We show an overview of our method in Fig. 3. Given a noisy latent, $\mathbf{z}_t$, and text prompts $y_v$ and $y_a$ corresponding to the desired image and spectrogram prompt respectively, we compute two CFG noise estimates (Eq. (2)):

$$\boldsymbol{\epsilon}_v^{(t)} = \boldsymbol{\epsilon}_{\phi,v}(\mathbf{z}_t; \varnothing, t) + \gamma_v \left( \boldsymbol{\epsilon}_{\phi,v}(\mathbf{z}_t; y_v, t) - \boldsymbol{\epsilon}_{\phi,v}(\mathbf{z}_t; \varnothing, t) \right), \tag{3}$$

$$\boldsymbol{\epsilon}_a^{(t)} = \boldsymbol{\epsilon}_{\phi,a}(\mathbf{z}_t; \varnothing, t) + \gamma_a \left( \boldsymbol{\epsilon}_{\phi,a}(\mathbf{z}_t; y_a, t) - \boldsymbol{\epsilon}_{\phi,a}(\mathbf{z}_t; \varnothing, t) \right), \tag{4}$$

where $\gamma_v$ and $\gamma_a$ are the corresponding visual and audio guidance scales. We then combine the noise estimates from both modalities by applying weighted averaging, producing a multimodal noise estimate that steers the denoising process toward a sample that is likely under the distribution of both images and spectrograms:

$$\tilde{\boldsymbol{\epsilon}}^{(t)} = \lambda_a^{(t)} \boldsymbol{\epsilon}_a^{(t)} + \lambda_v^{(t)} \boldsymbol{\epsilon}_v^{(t)}, \tag{5}$$

where $\lambda_a^{(t)}$ and $\lambda_v^{(t)}$ are the weights of the audio and visual noise estimates at timestep $t$ respectively.

With this new noise estimate $\tilde{\boldsymbol{\epsilon}}^{(t)}$, we perform a step of DDIM (Eq. (1)) to obtain a less noisy latent, $\mathbf{z}_{t-1}$. Repeating this process we obtain the clean latent $\mathbf{z}_0$, which is then decoded using the decoder $\mathcal{D}$ to obtain the spectrogram $\hat{\mathbf{x}} = \mathcal{D}(\mathbf{z}_0)$. This spectrogram can further be converted to a waveform using a pretrained vocoder or colorized to an RGB image whose grayscale version is the spectrogram.

**Warm-starting.** We find it useful to warm-start the denoising process. In Sec. 4.5, we experiment with warm-starting using only the spectrogram noise estimates or only the image noise estimates. This can be represented by using $w_a^{(t)}$ and $w_v^{(t)}$ as the relative weight on the audio and the visual noise estimates respectively. We let

$$\lambda_a^{(t)} = \frac{w_a^{(t)}}{w_a^{(t)} + w_v^{(t)}}, \quad \lambda_v^{(t)} = \frac{w_v^{(t)}}{w_a^{(t)} + w_v^{(t)}}, \tag{6}$$

with $w_a^{(t)} = H(t_a T - t)$ and $w_v^{(t)} = H(t_v T - t)$ being Heaviside step functions, and $t_a$ and $t_v$ indicating the *proportion* of the reverse process that has audio or visual denoising respectively. When $t_a < 1.0$ and $t_v = 1.0$, we warm-start with only image denoising, and vice-versa. The above ensures that the weights $\lambda_a^{(t)}$ and $\lambda_v^{(t)}$ sum to one, and are equally weighted after warm-starting.

**Colorization.** After we generate a spectrogram, $\hat{\mathbf{x}}$, we can optionally colorize it to create a more visually appealing result. Since our spectrograms fall outside the distribution of pre-trained colorization models, we use Factorized Diffusion [41] to colorize, which samples a diffusion model while projecting the noisy intermediate images such that they equal $\hat{\mathbf{x}}$ when turned into grayscale. In doing so, the denoising process synthesizes only the "color component" of the sampled image, while the "grayscale component" is constrained to equal the generated spectrogram. Note that this method is similar to prior work [63, 23, 106, 114]. We choose this particular method due to its simplicity.

## 4 Experiments

We evaluate our methods using quantitative metrics and human studies. We also present qualitative comparisons and an analysis of our method, and why it works.

### 4.1 Implementation Details

**Models.** We select a pair of off-the-shelf latent diffusion image and audio models that share the same latent space, encoder, and decoder. For the image model, we use Stable Diffusion v1.5[3] [96]. For the audio model, we use Auffusion[4] [118], which finetunes Stable Diffusion v1.5 on log-mel spectrograms. To synthesize audio from the log-mel spectrograms, we consider two options: following [118] and using off-the-shelf HiFi-GAN [66] vocoder, or the Griffin-Lim algorithm [47, 92]. We use HiFi-GAN for our main experiments. In Sec. 4.5, we evaluate the choice of vocoder and verify that our resultant waveforms do indeed encode to a visually interpretable spectrogram.

---

[3]Stable Diffusion v1-5 hugging face model card    [4]Auffusion hugging face model card

Table 1: **Quantitative evaluation on images that sound.** We report CLIP, CLAP, FID, and FAD metrics, along with 95% confidence intervals shown in gray. The best results are highlighted in **bold**.

| Method | Modality | CLIP (%) ↑ | CLAP (%) ↑ | FID ↓ | FAD ↓ |
|---|---|---|---|---|---|
| Stable Diffusion [96] | $\mathcal{V}$ | **34.5** $(\pm 0.1)$ | 2.2 $(\pm 0.2)$ | – | 41.74 |
| Auffusion [118] | $\mathcal{A}$ | 22.5 $(\pm 0.1)$ | **48.3** $(\pm 0.6)$ | 290.29 | – |
| Imprint | $\mathcal{A}$ & $\mathcal{V}$ | 27.2 $(\pm 0.2)$ | 32.3 $(\pm 1.0)$ | 244.84 | 29.42 |
| SDS | $\mathcal{A}$ & $\mathcal{V}$ | 25.4 $(\pm 0.2)$ | 23.4 $(\pm 1.4)$ | 273.03 | 32.57 |
| Ours | $\mathcal{A}$ & $\mathcal{V}$ | **28.2** $(\pm 0.1)$ | **33.5** $(\pm 0.9)$ | **226.46** | **19.21** |

**Hyperparameters.** We begin the reverse process with random latent noise $\mathbf{z}_T \in \mathcal{R}^{4\times32\times128}$, the same shape that Auffusion was trained on. Despite the image model not being trained on this specific size, we found that it nevertheless produces visually appealing results. We set the classifier guidance scales $\gamma_v$ and $\gamma_a$ to be between 7.5 and 10 and denoise the latents for 100 inference steps with warm-start parameters of $t_a = 1.0, t_v = 0.9$ to preserve audio priors. We decode the latent variables into images of dimension $3 \times 256 \times 1024$. By averaging across each channel, we obtain spectrograms corresponding to 10 seconds of audio. We re-normalize the spectrograms for visualization.

**Baselines.** As there is no previous work in this domain, we propose two baseline approaches. The first, inspired by Diffusion Illusions of Burgert *et al.* [11], uses multimodal score distillation sampling (SDS). We optimize a single-channel image $\mathbf{x} = g(\theta)$, where $g$ is an implicit function parameterized by $\theta$, using two SDS losses: one from the image diffusion model $\phi_v$ and the other from the audio diffusion model $\phi_a$. This results in a gradient of:

$$\nabla_\theta \mathcal{L}_{\text{SDS}}\left(\mathbf{x} = g(\theta)\right) = \lambda_{\text{sds}} \mathbb{E}_{t,\epsilon} \left[ \omega_v(t) \left( \boldsymbol{\epsilon}_v^{(t)} - \epsilon \right) \frac{\partial \mathbf{x}}{\partial \theta} \right] + \mathbb{E}_{t,\epsilon} \left[ \omega_a(t) \left( \boldsymbol{\epsilon}_a^{(t)} - \epsilon \right) \frac{\partial \mathbf{x}}{\partial \theta} \right], \quad (7)$$

where $\lambda_{\text{sds}}$ is the weight of the image SDS gradient and $\epsilon$ is the noise added to the image or latents. We implement this with pixel-based diffusion model DeepFloyd IF [26], as we find it performs better than Stable Diffusion with the SDS loss, and Auffusion [118]. This model thus does not require a shared latent space between vision and audio. We refer to this baseline as the *SDS*.

The second baseline involves taking existing images and subtracting them from existing spectrograms, multiplied by some scaling factor, inspired by [25]. This works when the spectrograms have high power, as the subtraction does not significantly affect the audio but still imprints an image into the spectrogram. We obtain spectrograms and images for this baseline via Auffusion and Stable Diffusion. This approach, which we call *imprint*, is simple but can be surprisingly effective. All methods use the same vocoder and post-processing for fairness. Please see Appendix A.3 for more details.

## 4.2 Quantitative Evaluation

We start by quantitatively evaluating the quality of our generated *images that sound*, examining how well the generated examples match the provided text prompts for each modality.

**Experimental setup.** Following the evaluation of Visual Anagrams [42], we create two sets of text prompt pairs. We randomly select 5 discrete (onset-based) and 5 continuous sound category names from VGGSound Common [17] as audio prompts. We randomly chose 5 objects and 5 scene classes for image prompts, formatted as "`a painting of [class], grayscale`". This yields a total of 100 prompt pairs. We report Stable Diffusion and Auffusion performances as single-modality

Table 2: **Human study.** We show win-rates of our spectrograms against those generated by the SDS and *imprint* baselines. The first row indicates which audiovisual prompt pair is evaluated, formatted as [audio prompt]/[visual prompt], with the last column being the average of all seven prompt pairs. Note that 50% win-rate is chance performance, and as such our method outperforms the baselines in the vast majority of cases. Also note that this is a *best-case* evaluation – please see Sec. 4.3 for details. All results reported are % win-rate against the baseline with a 95% confidence interval in gray ($N = 100$).

| Baseline | Metric | bell/castle | bark/dog | birds/garden | meow/kitten | racecar/racecar | tiger/tiger | train/train | Average |
|---|---|---|---|---|---|---|---|---|---|
| SDS | audio quality | 53.1 $(\pm 4.9)$ | 69.4 $(\pm 4.2)$ | 95.9 $(\pm 0.8)$ | 75.5 $(\pm 3.7)$ | 88.8 $(\pm 2.0)$ | 70.4 $(\pm 4.1)$ | 88.8 $(\pm 2.0)$ | **77.4** $(\pm 0.7)$ |
| | visual quality | 60.2 $(\pm 4.7)$ | 51.0 $(\pm 4.9)$ | 98.0 $(\pm 0.4)$ | 32.7 $(\pm 4.4)$ | 69.4 $(\pm 4.2)$ | 68.4 $(\pm 4.3)$ | 94.9 $(\pm 1.0)$ | **67.8** $(\pm 0.8)$ |
| | alignment | 58.2 $(\pm 4.8)$ | 63.3 $(\pm 4.6)$ | 93.9 $(\pm 1.1)$ | 62.2 $(\pm 4.7)$ | 82.7 $(\pm 2.8)$ | 59.2 $(\pm 4.8)$ | 91.8 $(\pm 1.5)$ | **73.0** $(\pm 0.8)$ |
| Imprint | audio quality | 82.1 $(\pm 3.0)$ | 73.7 $(\pm 3.9)$ | 53.7 $(\pm 5.0)$ | 54.7 $(\pm 5.0)$ | 86.3 $(\pm 2.4)$ | 85.3 $(\pm 2.5)$ | 85.3 $(\pm 2.5)$ | **74.4** $(\pm 0.7)$ |
| | visual quality | 92.6 $(\pm 1.4)$ | 86.3 $(\pm 2.4)$ | 66.3 $(\pm 4.5)$ | 68.4 $(\pm 4.3)$ | 66.3 $(\pm 4.5)$ | 77.9 $(\pm 3.5)$ | 56.8 $(\pm 4.9)$ | **73.5** $(\pm 0.8)$ |
| | alignment | 88.4 $(\pm 2.1)$ | 87.4 $(\pm 2.2)$ | 60.0 $(\pm 4.8)$ | 65.3 $(\pm 4.6)$ | 86.3 $(\pm 2.4)$ | 80.0 $(\pm 3.2)$ | 85.3 $(\pm 2.5)$ | **78.9** $(\pm 0.6)$ |

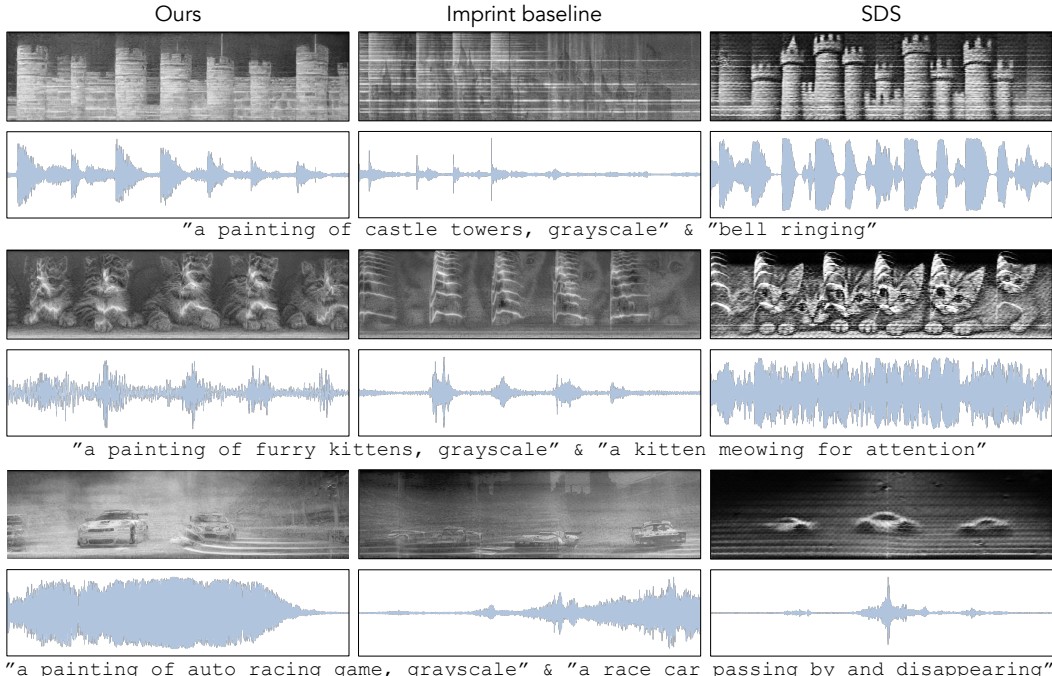

Figure 4: **Qualitative comparison.** We show our qualitative results along with the *imprint* and SDS baselines given visual (first) and audio (second) prompts. Please zoom in for better viewing.

benchmarks to establish upper and lower bounds. We generate 10 samples for each prompt pair, except for the SDS baseline, for which we generate 4 samples due to its slower speed.

**Evaluation metric.**    Following [50], we use the CLIP [95] score to measure the alignment between spectrograms and image text prompts, and analogously we use the CLAP [116] score to evaluate the alignment of audio with audio text prompts. An ideal method should excel at both simultaneously. We also report FID [51] and FAD [65] to evaluate the quality of generated examples where we use the results of Stable diffusion and Auffusion as reference sets respectively.

**Results.**    We show our quantitative results in Tab. 1. Our method outperforms baselines across all metrics and performs comparably to single-modality models, which serve as rough upper bounds for each modality. This demonstrates our approach's ability to generate meaningful *images that sound*, sampling from the intersection of natural image and spectrogram distributions. Stable Diffusion achieves a low CLAP score, indicating how poorly a randomly sampled natural image acts as a spectrogram. We observe that the SDS baseline often fails to optimize both modalities together. In contrast, our method achieves a higher success rate and generates more diverse results. Our method is significantly faster, generating one sample in 10 seconds compared to the SDS baseline's 2-hour optimization time using NVIDIA L40s. The *imprint* baseline imprints the image onto the spectrogram, potentially degrading the sound pattern and leading to a lower CLAP score. Note that FID and FAD are distribution-based metrics, and as our task focuses on generating examples that lie in a small subset of the natural image and spectrogram distribution, higher FID scores, in general, are expected.

## 4.3 Human Studies

**Experimental setup.**    We also perform two-alternative forced choice (2AFC) studies to evaluate our results. We construct seven paired text prompts by hand, ensuring semantic correlations between image and audio prompts, such as pairing a visual of dogs with the sound of dogs barking. Using these prompts, we generate samples using our method, the SDS baseline, and the *imprint* baseline, and hand-pick the best examples for evaluation. This *best-case evaluation* is useful as participants from MTurk are not expected to have prior knowledge about spectrograms, let alone domain expertise. Moreover, this evaluation matches the intended use case of our method, in which a user repeatedly queries the model for a result that they prefer based on artistic merit and quality. Participants, are presented with one sample from our method, and a corresponding sample from a baseline, and are

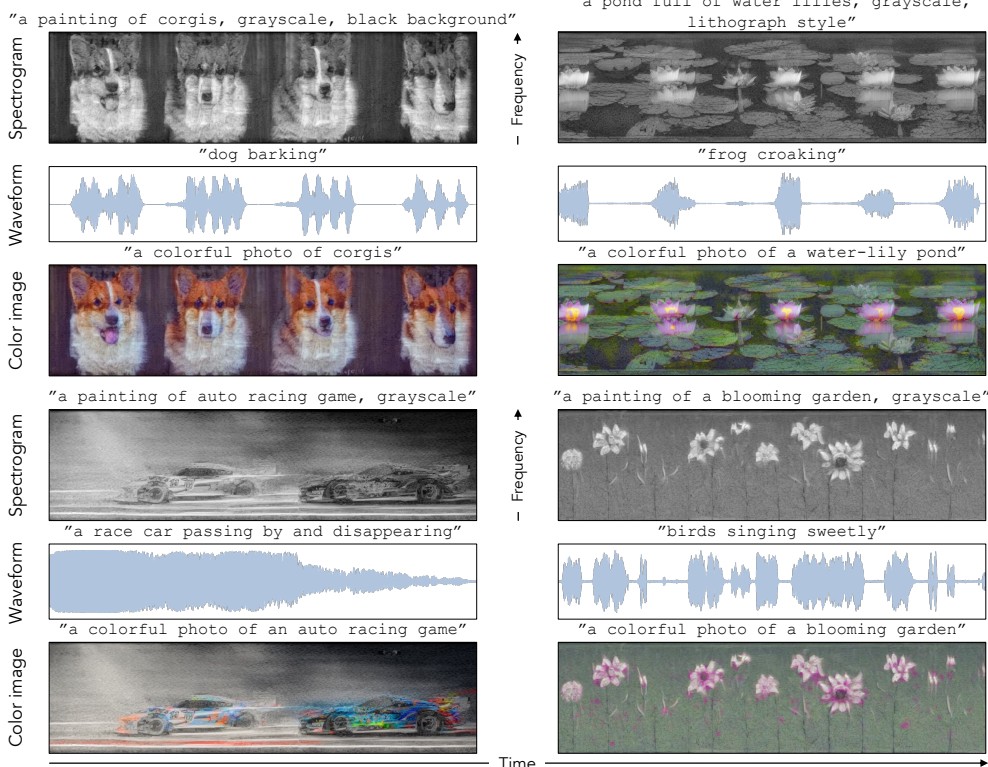

Figure 5: **Qualitative examples with colorization results.** We present 4 examples alongside their image prompts, audio prompts, and colorization prompts. Please refer to our website for video results.

asked to choose (1) the sample that looks most like the image prompt, (2) the sample that sounds most like the audio prompt, and (3) the sample in which the visual structure of spectrogram best aligns with that of image over time. The first two questions act as perceptual versions of CLIP and CLAP scores. The third question is designed to evaluate how well the visual structure of audio matches with the images as the spectrogram is played. Please see Appendix A.3 for further details and discussion.

**Results.**    Win-rates between our method and baselines are presented in Tab. 2, broken down by prompt pair. We also include averaged win-rates over all prompt pairs in the final column. As can be seen, our method outperforms the two baselines in most cases. Human evaluators consistently rate our spectrograms as being higher in audio and visual quality, and as being better "visually-synced"; on average our method is 2-3 times as likely to be chosen as the better sample than baselines.

### 4.4   Qualitative Results

**Results.**    We present qualitative results from our method as well as baselines in Fig. 4, with additional results from our method in Figs. 5, 8 and 9 in the appendix. Audio of all results can be found on our website. As can be seen (and heard) our approach generates more visually appealing samples with better sound quality than compared to the baselines. The SDS baseline often focuses on one modality, to the detriment of the other, and in general, generates audio of lower quality. Moreover, the method suffers from the characteristic oversaturation of SDS-based results. The performance of the *imprint* baseline is highly dependent on the independently generated spectrogram and image and tends to fail when the two are misaligned, as in the castle example, or when the spectrogram has low energy, as the subtracted image is hard to see in already low energy regions of the spectrogram, such as with the kitten example. Interestingly, we found that our method often combines visual and acoustic elements. For example, the onsets of the bells ringing in Fig. 4 coincide with the towers of the castle, and the spectrogram patterns of meowing are hidden as stripes and edges on the kittens.

We show additional hand-picked results from our approach in Fig. 5 with colorization results, in which we can see more examples of our method blending acoustic and visual elements, such as the

Table 3: **Ablations.** We conduct the ablation study of the guidance scale (left) and the warm-starting (right). To evaluate overall performance, we normalize each score by boundaries in Tab. 1 and then sum them.

| Method | Variation | CLIP (%) ↑ | CLAP (%) ↑ |
|---|---|---|---|
| | $\gamma_v, \gamma_a = 5.0$ | 28.0 | 29.3 |
| Ours | $\gamma_v, \gamma_a = 7.5$ | **28.2** | 31.9 |
| | $\gamma_v, \gamma_a = 10$ | **28.2** | **33.5** |

| Method | $t_v$ | $t_a$ | CLIP (%) ↑ | CLAP (%) ↑ | Overall ↑ |
|---|---|---|---|---|---|
| | 1.0 | 1.0 | 28.7 | 30.8 | 1.14 |
| Ours | 1.0 | 0.9 | **29.0** | 27.4 | 1.08 |
| | 0.9 | 1.0 | 28.2 | 33.5 | **1.15** |
| | 0.8 | 1.0 | 27.4 | **35.9** | 1.14 |

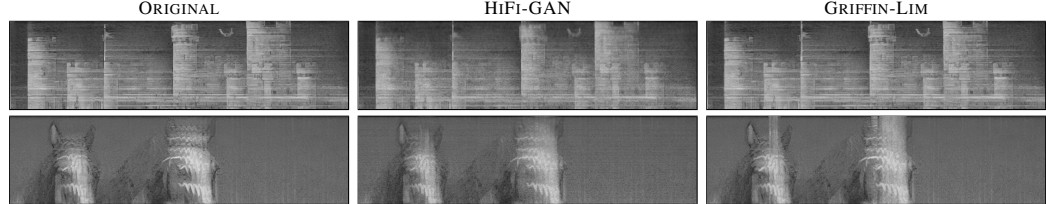

Figure 6: **Cycle consistency check on the vocoders.** We show the original log mel-spectrogram decoded from latents and log mel-spectrograms obtained from waveforms synthesized by HiFi-GAN or Griffin-Lim.

water lilies corresponding to the frogs croaking, the corgis corresponding to the dogs barking, and the flowers corresponding to the birds chirping. Please see more results in Fig. 8 of Appendix A.2.

**Multimodal compositionality.** Prior work [72, 29, 7, 42, 41] shows that diffusion models may be "composed" to generate samples that are likely under two or more different probability distributions. Our method can be seen as extending this idea of compositionality to multiple modalities. On the face of it, the distribution of spectrograms and the distribution of natural images would seem to be completely disjoint. However, as our results show, perhaps surprisingly, some overlap exists. We believe that this is possible for two reasons. First, spectrograms and images are both fairly flexible, allowing for significant amounts of perturbation or changes in style before becoming unrecognizable. And second, images and spectrograms share certain low-level characteristics, such as edges, curves, and corners, indicating a certain amount of similarity. Please see Appendix A.1 for more analysis.

However, we find that not all compositions can be successful as shown in Fig. 9 of Appendix A.2. Moreover, careful selection of prompts is crucial to creating good results. For example, incorporating terms such as "`lithograph style`" or "`black background`" encourages the visual model to create areas of silence, which results in better quality, as shown in Fig. 5.

### 4.5 Ablations

**Vocoder.** To extract waveforms from our generated spectrograms we use HiFi-GAN [66], a neural vocoder. Given that our spectrograms are incredibly out of distribution, one concern with this setup is that the vocoder will ignore spectrograms and generate waveforms that do not match the inputs. To ameliorate this concern, we conduct a cycle consistency check by re-encoding the neural vocoder's predicted waveform back into a spectrogram by performing an STFT. As can be seen in Fig. 6, the recomputed spectrograms are very similar to the original spectrogram, with only slightly less sharpness and some blurred textures[5]. This suggests we truly create visual spectrograms that look like images. We also experiment with using Griffin-Lim [92] as a vocoder, with similar results to HiFi-GAN as shown in Fig. 6. We opt to use HiFi-GAN as our default vocoder as it outperforms Griffin-Lim in audio quality, with Griffin-Lim attaining a CLAP score of **0.302**, compared to **0.335** obtained from HiFi-GAN. Please see more results in Fig. 7 of Appendix A.2.

**Warm-starting.** We also conduct an ablation study on our warm-starting strategy by varying which modality is warm-started and by how many steps. Results are presented in Tab. 3, where $t_a$ and $t_v$ are defined in Sec. 3.2. We find that warm-starting the denoising process with either image or audio diffusion yields higher scores in the corresponding modality, as that modality effectively gets free reign to set the high-level features of the final result. We find that allowing the audio diffusion model to denoise alone for the first 10% of the timesteps results in an attractive balance between CLIP and CLAP scores. Therefore, we adopt $t_v = 0.9$ and $t_a = 1.0$ for our main experiments.

---

[5]We note that perfect cycle consistency is not generally possible since vocoders are fundamentally lossy.

**Guidance scale.** We also explore different guidance scales $\gamma_v$ and $\gamma_a$ for our method. We present results in Tab. 3. We find that higher guidance scales generally yield better results on both modalities. We hypothesize that the higher guidance scales more strongly encourage the sample to come from the "intersection" of the conditional spectrogram and conditional image distributions, resulting in better alignment with both text prompts.

## 5 Discussion and Conclusion

In this work, we demonstrate that, perhaps surprisingly, there is a non-trivial overlap between the distribution of natural images and the distribution of natural spectrograms. We show this by sampling from the intersection of these two distributions, resulting in spectrograms that look like real images but also sound like real sounds. The method we proposed is simple and zero-shot, and leverages the compositional nature of diffusion with cross-modal models. We see our work as advancing multimodal compositional generation and opening up new possibilities for multimodal art.

**Limitations.** One limitation of our method is that it cannot generate examples that have both high-fidelity audio and image. We show failure cases, which occur for many prompts, in Fig. 9. Some of these failures may be due to the strict constraints of the problem, since realistic examples may not always exist at the intersection of both distributions. Our method is also limited by the quality of the audio diffusion model, whose performance lags behind that of visual models.

**Potential negative societal impacts.** The image and audio generation models that our method leverages are becoming progressively more powerful, and care must be taken in their deployment. Moreover, our method could potentially be used for steganography, secretly embedding images within audio. This capability may be used for deception, and we believe it deserves further consideration.

**Acknowledgements.** We thank Ang Cao, Linyi Jin, Jeongsoo Park, Chris Donahue, Alexei Efros, Prem Seetharaman, Justin Salamon, Julie Zhu, and John Granzow for their helpful discussions. This project is supported by the Sony Research Award and Cisco Systems. Daniel is supported by the National Science Foundation Graduate Research Fellowship under Grant No. 1841052.

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

## A.1 Multimodal Compositionality Analysis

**Model capabilities.** Through our experiments, we observed that our method generally performs well with "continuous" sound events (*e.g.*, racing cars or train whistles) and simple visual prompts. Continuous sounds typically produce spectrograms with high energy distributed across time and frequencies, resulting in "white" spectrograms. This allows our model to effectively reduce sound energy, creating visual patterns that align with the audio.

Simple visual prompts with object or scene nouns provide the diffusion models with more flexibility during denoising, enabling sampling from the overlapping distributions of images and spectrograms. However, more complex prompts could push the models into highly constrained distributions where images that sound are less likely to be generated.

Generating discrete sounds (*e.g.*, dog barking or birds chirping) is more challenging due to their sparse energy distribution. In these cases, the models are more constrained, making it difficult to produce visual content with clear edges and structures aligned with sound onsets, leading to less satisfactory results sometimes.

Additionally, we emphasize that some prompt pairs may not have overlapping image and spectrogram distributions, making it impossible to create meaningful examples. For instance, combining the visual prompt starry night with the audio prompt playing guitar leads to a conflict, where the image modality tends toward a dark image, while the audio suggests a brighter one.

**Style words.** Visual diffusion models are capable of generating RGB images, but spectrograms are only one channel. We therefore use style words like "`grayscale`" or "`black background`" to nudge the image denoising process toward a distribution that matches the spectrograms. As suggested by the reviewer, we conducted ablation

Table 4: **Ablation on style words.**

| Method | CLIP ↑ | CLAP ↑ | FID ↓ | FAD ↓ |
|---|---|---|---|---|
| w/o "grayscale" | 28.1 | **33.7** | 237.12 | **18.09** |
| w/ "grayscale" | **28.2** | 33.5 | **226.46** | 19.21 |

experiments by removing the grayscale style word. The results are shown in Tab. 4. The model produces similar results, but (as expected) the image quality slightly decreases while the audio quality slightly improves.

## A.2 Qualitative results

**More qualitative results.** We show more qualitative results from our method with different prompts in Fig. 8. Please see our website for video results. We also provide random examples with random prompt pairs in Fig. 9 with the last two rows as failure cases. For the failure cases, we can see that they either have good audio quality but lose clear visual patterns (mountains/fireworks) or have clear visual appearances but noisy audio (dogs/trains).

**Vocoder analysis.** We show more examples of the vocoder cycle consistency experiment in Fig. 7. As can be seen, the spectrograms from HiFi-GAN are quite similar to the original ones decoded from latents, indicating that our method does not find adversarial examples against the vocoder, but truly does find spectrograms that look like images.

## A.3 Implementation Details

**Colorization.** We use DeepFloyd IF[6] [26] following Factorized Diffusion [41] for colorizing spectrograms. This technique colorizes a grayscale image by using a pretrained diffusion model zero-shot to generate the color component, and is similar to prior work such as [63, 23, 106, 114]. We use it due to its simplicity. We colorize spectrograms of size $1 \times 256 \times 1024$ by directly feeding these into the diffusion model, which we found produced reasonable results despite the fact that the model was not trained for this size. We use prompts of the form "`a colorful photo of [image prompt]`" and denoise for 30 steps with a guidance scale of 10. Additionally, we found that starting the denoising at step 7 of 30 gave better results, which we hypothesize works because it gives the model a stronger prior for what the structure of the image is than starting from pure noise.

---

[6]https://huggingface.co/DeepFloyd

**SDS baseline.** We follow Diffusion Illusions [11] to implement our SDS baseline with an implicit image representation. We use Fourier Features Networks [109] with a learnable MLP to generate images of size $1 \times 256 \times 1024$. We use stage I of DeepFloyd IF-M to perform image score distillation sampling. We randomly make eight overlapping $256 \times 256$ crops and resize them to $64 \times 64$ to compute the averaged image SDS loss with a guidance scale of 80. For the audio modality, we use Auffusion [118]. As Auffusion is a latent diffusion model, we encode the images into $4 \times 32 \times 128$ latents and perform the audio SDS loss with a guidance scale of 10, which we found gave the best performance in the audio-only generation. We set the weight of the image SDS loss $\lambda_{\text{sds}}$ to $0.4$ to ensure balanced optimization for both modalities. We use the AdamW optimizer with a learning rate of $10^{-4}$ and weight decay of $10^{-3}$, and optimize the Fourier Feature Network for 40,000 steps. We also apply the warm-start strategy to this method by optimizing the audio SDS loss only for the first 5,000 steps by setting $\lambda_{\text{sds}}$ to zero. We note this method does not require a shared latent codebook between image and audio diffusion models.

**Imprint baseline.** We begin by generating images, $\mathbf{x}_{\text{img}}$, and spectrograms, $\mathbf{x}_{\text{spec}}$, of size $256 \times 1024$ using Stable diffusion and Auffusion, respectively, both with a guidance scale of 7.5. Next, we use the generated images as masks by converting them into inverse grayscale images and scaling them by a factor $\rho$. This mask is then applied to the generated spectrogram to obtain the final result, given by $\mathbf{x}_{\text{spec}}(1 - \rho \operatorname{gray}(1 - \mathbf{x}_{\text{img}}))$. The hyperparameter $\rho$ controls the strength of energy reduction: larger values yield clearer visual patterns but poorer audio quality, and vice versa. To strike a good balance, we set $\rho = 0.5$. The imprint baseline takes 10 seconds to generate a sample on NVIDIA L40s.

**Algorithm 1** Pseudocode in a PyTorch-like style for the *imprint* baseline.

```
# get images and specs from LDMs
img = stable_diffusion(text_v)
spec = auffusion(text_a)
# reduce the energy give image masks
mask = 1 - rho * (1.0 - img.mean(0))
spec = mask * spec
audio = vocoder(spec)
```

**Prompt selection.** We present the image and audio prompts used for the quantitative evaluation in Tab. 5. We use 10 prompts for each modality, for a total of 100 prompt pairs.

Table 5: **Text prompts for the quantitative evaluation.**

| Image prompts | Audio prompts |
| --- | --- |
| a painting of castles, grayscale | dog barking |
| a painting of dogs, grayscale | cat meowing |
| a painting of kittens, grayscale | bird chirping, tweeting |
| a painting of tigers, grayscale | tiger growling |
| a painting of auto racing game, grayscale | church bell ringing |
| a painting of mountains, grayscale | race car, auto racing |
| a painting of a garden, grayscale | train whistling |
| a painting of a forest, grayscale | fireworks banging |
| a painting of a farm, grayscale | people cheering |
| a painting of a beach, grayscale | playing acoustic guitar |

## A.4 Human Studies

Participants for the human study were recruited from Amazon Mechanical Turk (MTurk), and were paid 0.50 USD for a task lasting less than 5 min. We use a total of seven prompt pairs and compare them against two baselines: the SDS baseline and the *imprint* baseline. For each method and prompt pair, we hand-selected two high-quality samples for a total of 84 videos. Each video is about 10 seconds long, and includes a vertical line moving from left to right, indicating the current temporal position in the spectrogram. All participants were shown 14 pairs of videos–seven pairs comparing our method to the SDS baseline, and seven pairs comparing our method to the *imprint* baseline, all randomly selected and blinded. The participants are then asked to answer three questions:

1. Which video LOOKS most like a `[visual prompt]`?
2. Which video SOUNDS most like a `[audio prompt]`?
3. In the video, we play the image as a sound, from left to right. In which video does the `[visual prompt]` better align with the `[audio prompt]` sounds?

| ORIGINAL | HIFI-GAN | GRIFFIN-LIM |
|----------|----------|-------------|

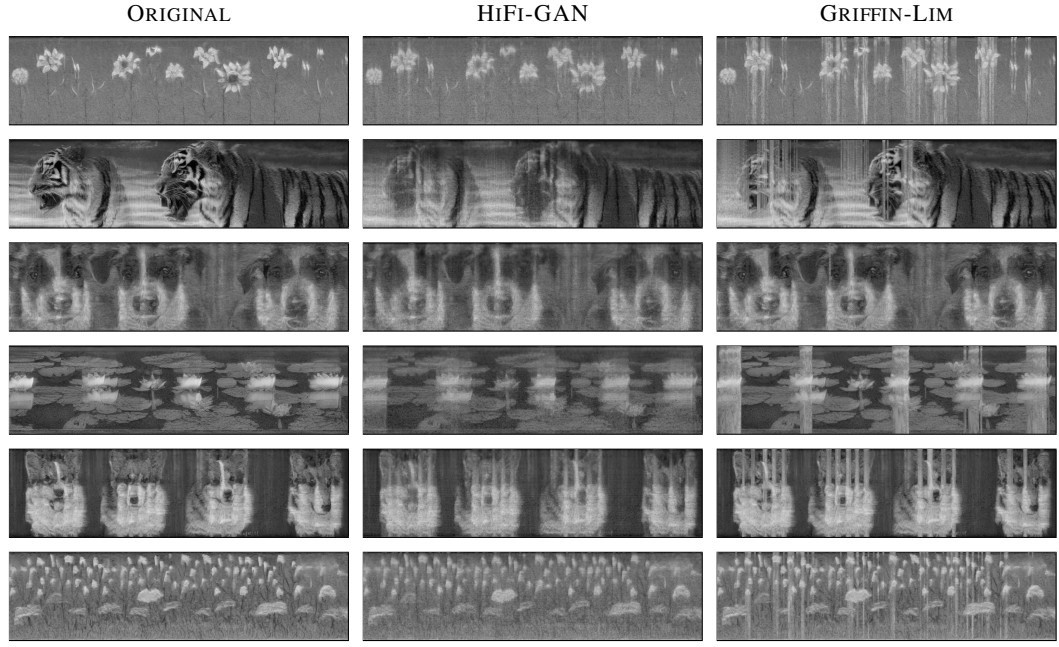

Figure 7: **More results on the vocoder cycle consistency check.** We show the original log mel-spectrogram decoded from latents and log mel-spectrograms obtained from waveforms synthesized by HiFi-GAN and Griffin-Lim.

The first two questions are designed to evaluate the quality of the audio and image generated by the methods, and their alignment with the respective prompts. The third question seeks to understand how well the visual structure or texture of the generated image and spectrogram align. However, note we were not able to guarantee that the participants had prior experience with spectrograms. To mitigate this to an extent, we include the description as a preamble to the third question. Also, note that we use abbreviated versions of the audio and visual prompts to avoid excessively long questions. We provide the prompt pairs we used for human studies in Tab. 6 for reference, and screenshots of our survey including the title block as well as the first video pair and associated questions in Fig. 10.

Table 6: **Text prompts for the human studies.** We note that the prompts are paired.

| Image prompts | Audio prompts |
|---------------|---------------|
| a painting of castle towers, grayscale | bell ringing |
| a painting of cute dogs, grayscale | dog barking |
| a painting of a blooming garden with many birds, grayscale | birds singing sweetly |
| a painting of furry kittens, grayscale | a kitten meowing for attention |
| a painting of auto racing game, grayscale | a race car passing by and disappearing |
| a painting of trains, grayscale | train whistling |
| a painting of tigers, grayscale | tiger growling |

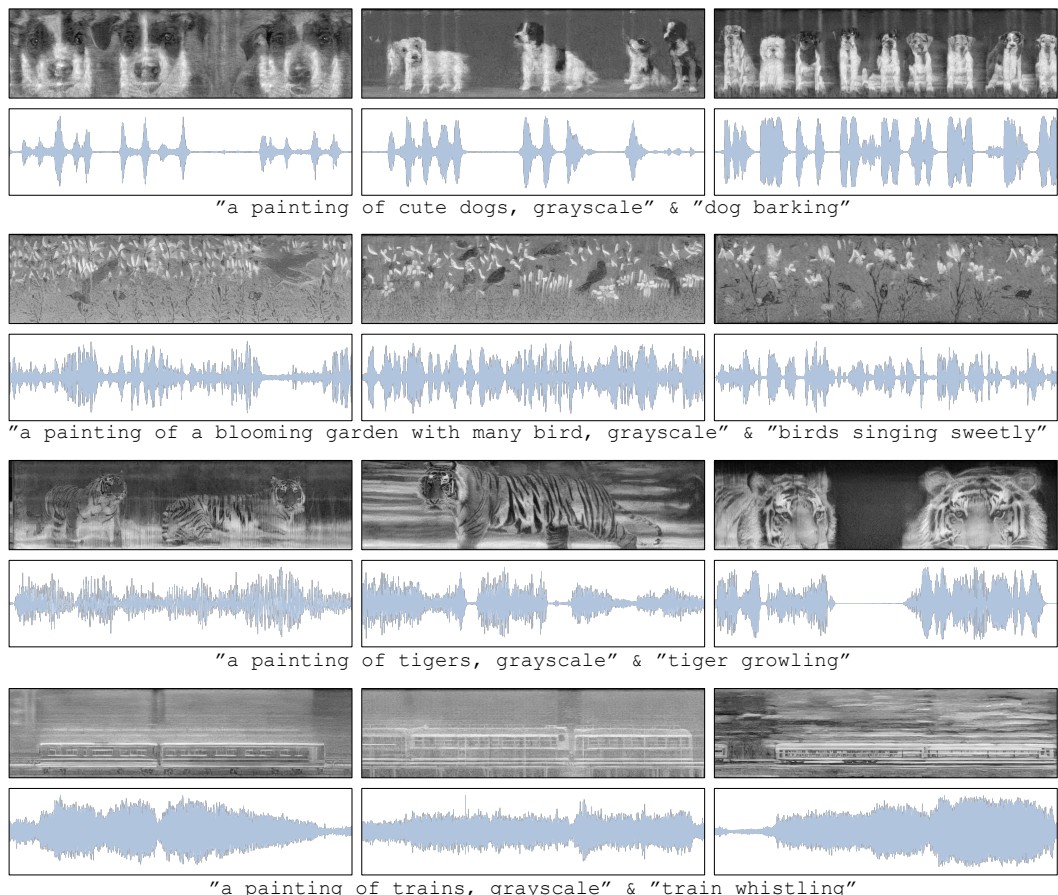

Figure 8: **More qualitative results.** We show more qualitative results of our approach. Please zoom in for better viewing.

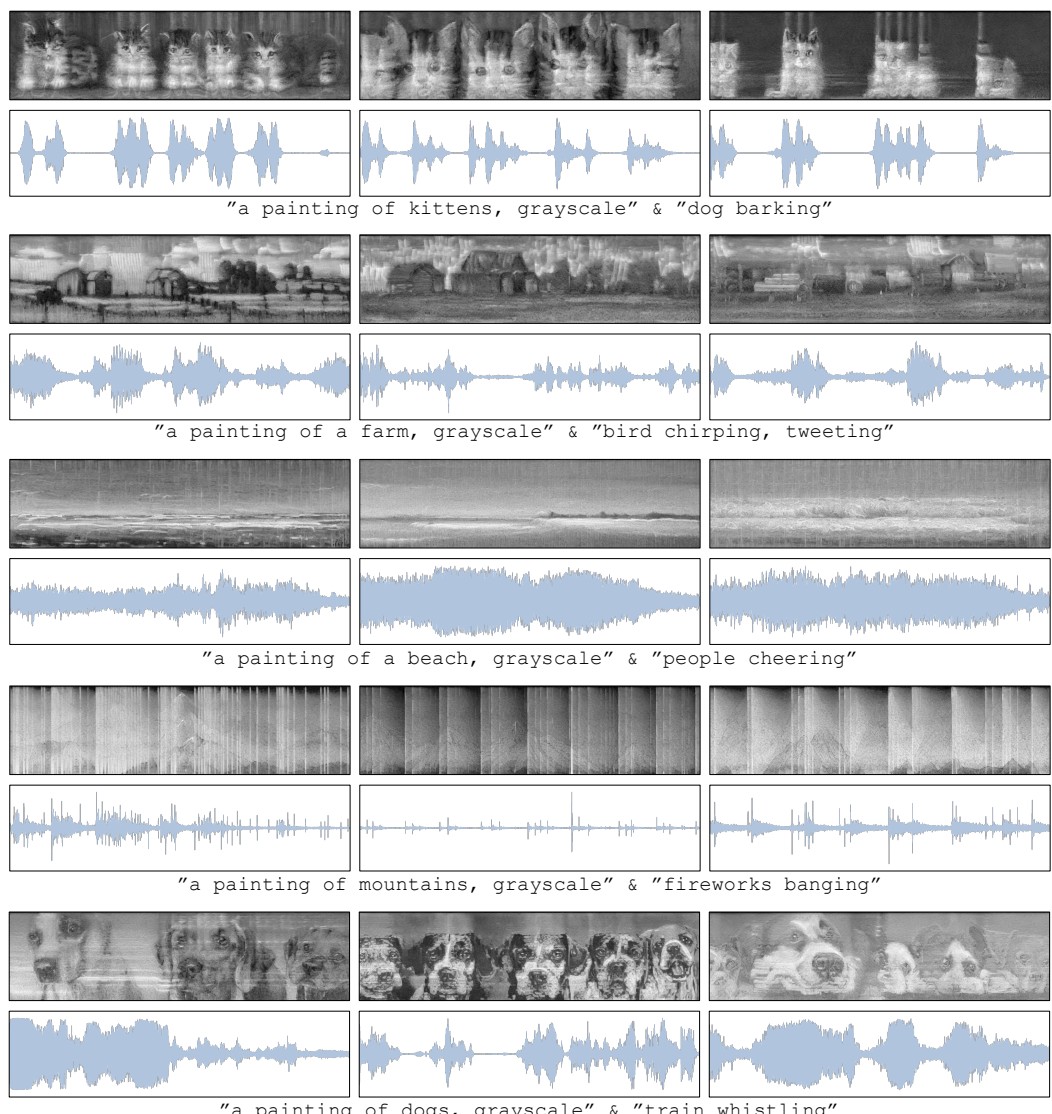

Figure 9: **Random results.** We show random results from our approach, using random audio and visual text prompt pairs. We provide failure cases in the last two rows. Please zoom in for better viewing.

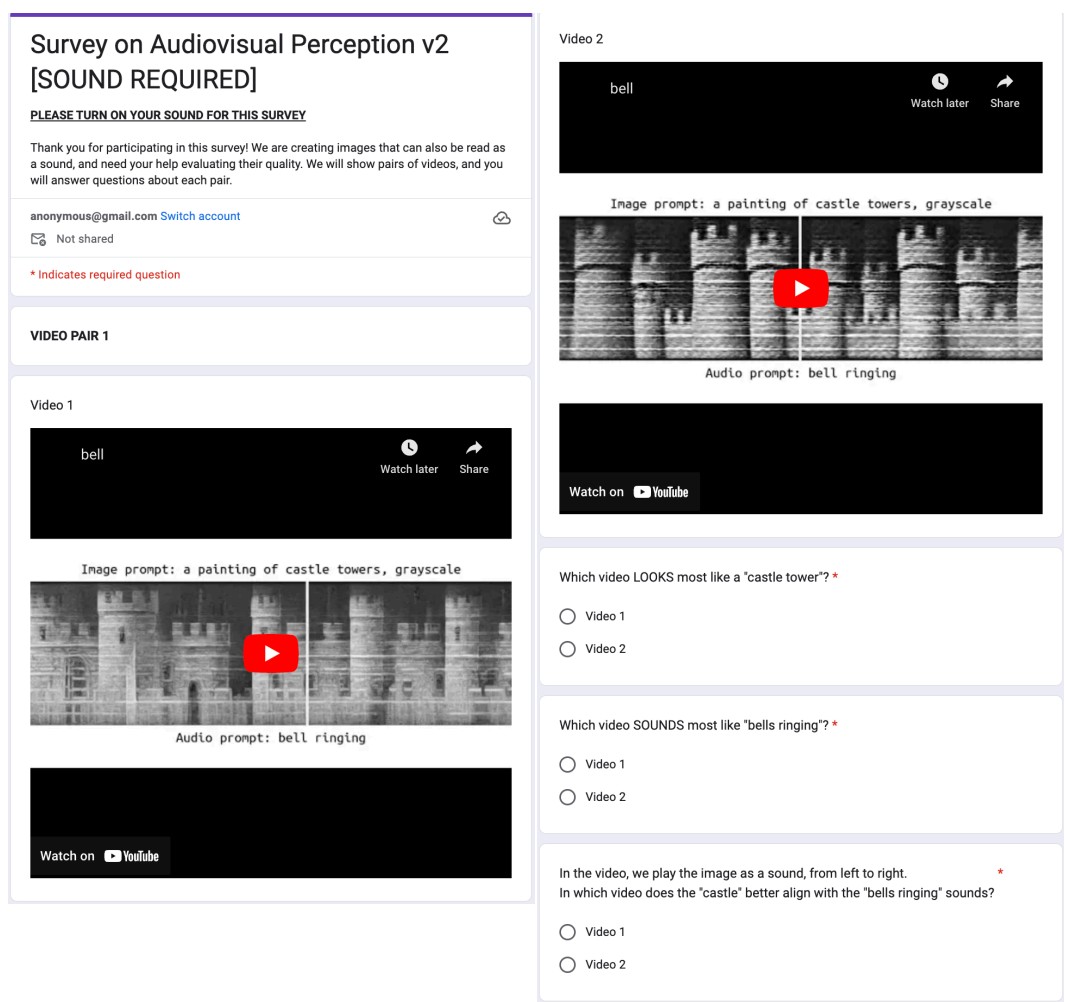

Figure 10: **Human study screenshots.** We show screenshots from our human study survey. Here we show the title block, as well as the first pair of videos. The full survey contains 14 video pairs.

