# OpenReview forum: "Images that Sound: Composing Images and Sounds on a Single Canvas"
_NeurIPS.cc/2024/Conference — NeurIPS 2024 poster_

### Official Review · Reviewer_w6xJ · 2024-07-07

**Soundness:** 3
**Presentation:** 3
**Contribution:** 2
**Rating:** 5
**Confidence:** 4

**Summary:**

This paper explores the feasibility of synthesizing spectrograms (images that sound) that simultaneously look like natural images and sound like natural audio. This paper proposes a zero-shot approach, which leverages pre-trained text-to-image and text-to-spectrogram diffusion models that operate in a shared latent space, and denoises noisy latents with both the audio and image diffusion models in parallel. This paper shows quantitative evaluations and perceptual studies to claim the proposed method is able to generate spectrograms that align with a desired audio prompt while also matching the corresponding visual appearance.

**Strengths:**

+ The topic introduced in this paper is very interesting and inspiring.
+ The paper is clearly written, providing detailed explanations for easy understanding and reimplementation. The proposed method's capacity is fairly claimed and supported by convincing experimental results.

**Weaknesses:**

- Technical contribution is limited. The proposed method combines two existing latent diffusion models (audio and visual), weighted integrate noises from these two processes to ensure semantic consistency for both modalities. Additionally, another existing diffusion model is used for colorization. The technical innovation appears incremental, raising concerns about the novelty of the contribution.
- The capability of the proposed model is not high. Despite the inspiring topic, the generated audio and visual quality show a noticeable performance gap compared to state-of-the-art generators. Ensuring consistency of both modalities in one image may result in less diverse outputs for audio and visual generation, potentially failing to accurately reflect prompt details.

**Questions:**

Please see the weaknesses section for my major concerns.

**Limitations:**

The authors have explained the current limitations, and potential negative social impact in the paper.

---

> ### Author Rebuttal · Authors · 2024-08-06
>
> We thank the reviewer for their valuable feedback. Below are our responses.
>
> &nbsp;
>
> **Technical contribution**
>
> The reviewer appears to have taken a narrow, algorithm-centric view of what constitutes a contribution. We note that this seems to be the major complaint in their (unusually short) review. Our work in fact makes a number of contributions. It is the first to propose the idea of generating examples that are natural in visual and audio signals, thereby exploring the intersection of two seemingly very different distributions. We are not aware of any work in the field of multimodal learning that has addressed this problem before. Beyond the novelty of this problem formulation (as acknowledged by mkQc, RECb, and quDM), the fact that we can find such examples is an interesting empirical contribution; it was not obvious *ex ante* that this could be done. Finally, we feel that our approach's simplicity is a major *strength*, not a drawback. While our approach draws inspiration from [38, 65], those approaches only combined noise from a single diffusion model. Showing that techniques from compositional generation can be adapted to this highly novel multimodal domain, and that (perhaps surprisingly) noise estimates from two different modalities' diffusion models can be successfully combined together, goes far beyond previous work.
>
> &nbsp;
>
> **Colorization**
>
> To clarify, we provide colorized versions of the spectrograms in several qualitative examples using a post-processing procedure, since spectrograms are limited to grayscale. This is neither part of the evaluation nor a key component of our main method.
>
> &nbsp;
>
> **Model capabilities**
>
> First, we want to stress that generating examples that can be both viewed as images and listened to as sound in a zero-shot manner is extremely challenging: it requires using a single signal to represent two very different modalities without any supervision. As noted in our limitations section, the nature of this task *forces* the model to produce examples that may not be as natural as those from specialized models in each individual modality, since many visual patterns are improbable under audio models, and vice versa. Interestingly, our model generates high-quality results in both modalities despite these constraints. We have also shown that our method significantly outperforms several challenging baselines and ablations, providing further evidence for its abilities.
>
> &nbsp;
>
> We hope these comments have addressed the reviewer’s questions. We would like to thank them for their time and ask that they consider raising their score for our paper.

---

> > ### Comment · Reviewer_w6xJ · 2024-08-11
> >
> > Thanks for the authors' response. However, I think I need to re-clarify my concerns after reading the response.
> > 1) Technical contribution. I'm a little bit confused about the definition of "narrow, algorithm-centric view" and "unusually short reviews". I agree with the novelty and contribution of bringing such an inspiring and interesting task to the community. The technical details, proposed solution and evaluations are all sound and fair. I don't have concerns or questions on the details of the work except for the overall questions. So I mentioned about my concern only on the "technical contribution" side, which specifically means the novelty in terms of bridging the existed methods is a simple way. Indeed, I agree that simplicity is an advantage. So I list this concern as one of the possible weaknesses for the reviewers and AC to judge thoroughly. Just to clarify, I will only consider this part as one of my concerns but not a critical issue of this work that blocks it from publishing.
> > 2) Colorization. This is not my major concern or even a question. I mentioned this because of Line 151 "we use Factorized Diffusion [37] to colorize", to support my concern on the technical contribution reviews.
> > 3) Model capabilities. The model capability is indeed my major concern. I understand the task is a challenging one, but sometimes we as reviewers can have different standard to balance the capacity and an interesting task for a conference paper. I acknowledge that the authors have shown their capacity and advantages on fair evaluation scope. But after reading the paper and response I still feel it would be a problem for the proposed method to scale on general audio-visual examples. A more strong response on this side with more detailed explanations on what case can generally work and what can't, or potential solutions on that as well as analysis etc. can all help solve my concerns.
> > Given the consideration above, I gave the borderline reject rating. I look forward to have more fair discussion on these side and will be very open to change my mind. It won't bother me if the paper gets accepted even with my current rating. Sorry for causing the misunderstanding from my previous reviews. I think this can better clarify my points.

---

> > > ### Author Response · Authors · 2024-08-12
> > >
> > > We thank the reviewer for the further clarification about their reviews. Below are our new responses.
> > >
> > > &nbsp;
> > >
> > > **Technical contribution**
> > >
> > > Our main method is simple but non-trivial for combining noise estimates from two different modalities' diffusion models, which no prior work has explored. Besides our main method, we also proposed two other methods called *imprint* and *SDS* baseline that generate *images that sound* examples from different perspectives (L174-188). Overall, we believe our work makes significant technical contributions. For clarification, colorization is not one of our claimed contributions. We simply use it to improve the visualization quality of the results.
> > >
> > > &nbsp;
> > >
> > >
> > > **Model capabilities**
> > >
> > > The reviewer states that their primary concern is a lack of analysis of the capabilities of our model. We discussed some of them in L257 and L293. Further, we give a detailed explanation below:
> > >
> > > First, we want to emphasize that the nature of this task inherently forces the model to produce examples that may not appear as natural as those from specialized models in each modality, as many visual patterns are unlikely to align with audio models. In other words, realistic examples are constrained to the overlapped distribution between both modalities, which inherently limits the quality of the generated results. Furthermore, our experiments show that we outperform many other possible approaches, highlighting the capability of our approach. Our results also qualitatively outperform artist-created examples, which often produce random noise (e.g., Aphex Twin and others [9]).
> > >
> > > Through our experiments, we observed that our method generally performs well with "continuous" sound events (e.g., racing cars or train whistles) and simple visual prompts. Continuous sounds typically produce spectrograms with high energy distributed across time and frequencies, resulting in “white” spectrograms. This allows our model to effectively reduce sound energy, creating visual patterns that align with the audio.
> > >
> > > Simple visual prompts with object or scene nouns provide the diffusion models with *more flexibility* during denoising, enabling sampling from the overlapping distributions of images and spectrograms. However, more complex prompts could push the models into highly constrained distributions where *images that sound* are less likely to be generated.
> > >
> > > Generating discrete sounds (e.g., dog barking or birds chirping) is more challenging due to their sparse energy distribution. In these cases, the models are more constrained, making it difficult to produce visual content with clear edges and structures aligned with sound onsets, leading to less satisfactory results **sometimes**.
> > >
> > > Additionally, we emphasize that some prompt pairs may not have overlapping image and spectrogram distributions, making it impossible to create meaningful examples. For instance, combining the visual prompt `starry night` with the audio prompt `playing guitar` leads to a conflict, where the image modality tends toward a dark image, while the audio suggests a brighter one.
> > >
> > > Lastly, we note that our use of off-the-shelf Stable Diffusion, trained on $512 \times 512$ RGB images, to generate $256 \times 1024$ grayscale images directly could potentially limit our model’s capabilities. The performance could be possibly improved by having a specific image diffusion model that is suitable for this task. We will include a more detailed analysis in the manuscript.
> > >
> > > &nbsp;
> > >
> > > We hope these comments have addressed the reviewer’s questions

---

> > > > ### Comment · Reviewer_w6xJ · 2024-08-14
> > > >
> > > > Thanks for the further clarification. These make sense and improve the interpretability of the proposed model. I suggest the authors could include these interesting observations and conclusions in the analysis. I have increased my rating.

---

> > > > > ### Author Response · Authors · 2024-08-14
> > > > >
> > > > > We thank the reviewer for recommending acceptance. We appreciate this discussion and will incorporate those changes in our final version.

---

### Official Review · Reviewer_quDM · 2024-07-12

**Soundness:** 3
**Presentation:** 3
**Contribution:** 2
**Rating:** 4
**Confidence:** 3

**Summary:**

The paper proposes using pretrained text-to-image and text-to-speech diffusion and leveraging their compositional property to generate spectrograms that look like images and can be also be converted into meaningful sounds. The work is motivated by applications in art. The paper also curates a set of text prompts for conditioning the diffusion models. Besides, it evaluates the generated spectrograms using both automatic and human evaluation. In automatic evaluation, it measures both intra-modal similarity between the generated output and the modality-specific text prompts using CLIP-style models. In human evaluation, human subjects are asked to rate the output compared to baselines' on the basis of both intra- and cross-modal similarity between the generated output and the modality-specific text prompts. Furthermore, to tackle the lack of baselines for the task, the paper introduces two baselines by adapting existing methods, and shows that the proposed method performs better than the baselines across different evaluation types and metrics. Finally, the paper also provides good qualitative examples that show the promise of the idea.

**Strengths:**

1. Interesting idea: the idea of leveraging pretrained diffusion models and their compositionality to render spectrograms that look like real images and sound like real sounds is interesting and could have useful applications in art, as mentioned in the paper

2. Extensive evaluation and good qualitative results: the paper extensively evaluates its model using both automatic and human evaluation, different evaluation metrics, and compares against different baselines despite the lack of existing methods for the task. Besides, it does important ablations of its method, which helps better understand the role of different design choices. Finally, the paper provides good qualitative examples, which help further demonstrate the strengths of the idea.

3. Useful baselines: to tackle the lack of existing methods, the paper proposes two meaningful baselines for the task, which not only facilitate better model evaluation, but can potentially be useful for future work on this topic

**Weaknesses:**

1. Text is unclear / poorly structured at parts:
    i) L21-23, "We hypothesize ... share statistical properties ...readily process": what kind of statistical properties is the paper referring to? Clear examples in the next version could help a reader.
    ii) It's probably better to put the application/motivation para (L29-37) before the para in L21-28. The current order leaves a reader guessing the use of the work for quite a bit (at least that was the case with me)

2. Importance of shared latents (L72 and elsewhere): I am not entirely convinced by the current content of the text why a model that does not share latents won't work. There is no model analysis/ablation for this claim as well. I think that if the model is able to work even with different prompts for the text-to-image and  text-to-audio model, it might work with separate latent space as well.

3. Quantitative metrics are meaningful? if CLIP and CLAP are given equal weightage in table 1, auffusion is the strongest model, but obviously it isn't. This makes me wonder if these standalone quantitative metrics are even meaningful for this task.

4. Lack of other standard quantiative metrics: the paper does not evaluate the generated outputs using standard metrics [1, 2] like FID, inception score, etc.

5. Human study setup and results are not entirely clear.
   i) In L208-9, the paper says that most of the time the SDS baseline collapses to either modality. If that is true, how come the win rate against SDS is so much higher than 50% for both audio and visual quality?
   ii) Why are the examples hand-picked (L218) for human evaluation, shouldn't they be randomly sampled instead?
   iii) It's not clear how the cross-modality alignment evaluation (L222-3, and upper and lower last row in table 2) makes sense when the text prompts to the two diffustion models are independent

6. L278, "attractive balance between CLIP and CLAP scores": the choice of weightage on CLIP and CLAP for determining t_a and t_v in table 3 seems arbitrary. If they are given equal weightage, the best values would be t_v = 0.8 and t_a = 1.0?

7. What's the rationale behind limiting the prompts to the ones listed in table 4 given that the individual diffusion models work with a much broader set of prompts?

8. Minor:
   i) L264, "simply found adversarial examples against the vocoder": the meaning of this phrase was unclear to me

Refs:
[1] High-resolution image synthesis with latent diffusion models. Rombach et al.  CVPR 2022.
[2] Auffusion: Leveraging the power of diffusion and large language models for text-to-audio generation. Arxiv 2024.

**Questions:**

Could the rebuttal comment (more) on the following
1. What kind of statistical properties is the paper referring to in L21-23: "We hypothesize ... share statistical properties ...readily process"?

2. importance of shared latents (L72 and elsewhere) and if it is possible to provide an analysis/ablation to support the claim. See weakness 2 for details.

3. how meaningful the quantiative metrics are (see weakness 3 for details) and if it is possible to report other standard image and audio generation metrics (see weakness 4 for details)

4. i) why the model wins against SDS by a large margin even when "SDS baseline often fails to optimize both modalities together, producing either spectrogram or image content only" (L208-9). See weakness 5i for details.
    ii) why the samples for human evaluation are cherrypicked. See weakness 5ii for details.
    iii) why the cross-modality alignment metric makes sense when the prompts to the two diffusion models are not shared. See weakness 5iii for details.

5. how the CLIP and CLAP scores are weighted while determining which model is better (details in weakness 6 and also related to Q3)

6. the rationale behind limiting the prompts to the ones listed in table 4 given that the individual diffusion models work with a much broader set of prompts

**Limitations:**

The paper has discussed its limitations and societal impact.

---

> ### Author Rebuttal · Authors · 2024-08-06
>
> We thank the reviewer for their comprehensive feedback. Below are our responses.
>
> &nbsp;
>
> **Importance of shared latent spaces**
>
> This seems to be a misunderstanding: the latent spaces *must* be shared during joint sampling because if they weren't, then the latent vector would be decoded to two completely different images in the two modalities, which would trivialize into a different problem. Moreover, diffusion would fail if the latents did not span the same space or if the noise schedule differed. Finally, we stress our paper *does* include a model that works even without a shared latent space. The SDS version of the model (L174-182) successfully avoids this constraint because it directly backpropagates into pixels. It uses a pixel diffusion method (DeepFloyd) and an audio latent diffusion model (Auffusion). While it is outperformed by our joint sampling approach, this result suggests that the shared latent space is not strictly necessary. We will clarify this.
>
> &nbsp;
>
> **Writing of introduction**
>
> Statistical similarities between images and audio have been well established in previous work [A, B], which we would be happy to cite for clarity. However, we note that we have already provided several examples in the introduction: 1) we point out that both spectrogram and images have similar patterns that are famous "objects of study" in the natural image statistics (e.g., lines and edges) (L22-23), 2) the fact that frozen visual features have been surprisingly successful for audio models (L18-20). If there were no common statistical similarities between the two signals, then neither property would hold. Regarding the presentation order, we feel that the image statistics motivation helps readers understand the importance of the problem and motivates the generative modeling approach, which is why we described it before the task itself. We note that Fig.1 and its caption already present the work in a style that is quite similar to the reviewer's suggestion.
>
> &nbsp;
>
> **Quantitative metric**
>
> We report Stable Diffusion and Auffusion as single-modality models, serving as upper and lower bounds for reference. CLIP and CLAP metrics are meaningful when compared with multimodal baselines, showing how well the generated results align with their respective prompts. These two networks have different common output ranges, so naively summing them without calibration would emphasize one score at the expense of the other. To fairly evaluate overall performance, we normalize each score based on the lower and upper bounds listed in Tab.1 and then sum them. As a result, in Tab.3, $t_v=0.9, t_a=1.0$ achieves the highest score of 1.154, while $t_v=0.8, t_a=1.0$ scores 1.139. For the results of FID and FAD metrics, please refer to the general response. These details will be included in the revised version.
>
> &nbsp;
>
> **SDS baseline**
>
> We note that *collapsing to a single modality* does not necessarily mean that that modality is generated perfectly. It simply means that the other modality is not generated at all. Therefore, despite the collapse, the result may still be of poor quality, especially considering that the method is still jointly optimizing two losses. Additionally, due to the limitations of the SDS loss [83], the generated images tend to be overly saturated, while the audio is often distorted. As a result, the outcomes are generally inferior to our main method. For more examples, please refer to the supplementary video.
>
> &nbsp;
>
> **Human study**
>
> For the human study, we compare our 3 proposed models' ability to generate examples that are qualitatively and artistically appealing. For each method, we generate a fixed number of samples per prompt pair and hand-select the best qualitative result. This also avoids having the 2AFC task performance determined by a model's failure rate (e.g., SDS collapsing to one modality), which would dominate results. We note that the random results used in the quantitative evaluation can be found in Fig.8. We will clarify this in our manuscript.
>
> &nbsp;
>
> **Cross-modality alignment**
>
> We explain the cross-modality alignment metric in L623-629. This metric measures how well the visual structure of the image aligns with that of the spectrogram, rather than measuring *semantic* alignment, e.g., how an image edge of castle towers corresponds to the spectrogram onset pattern of ringing bells, or how an image of dogs matches the audio pattern of barking. Please see Fig.10 for the survey question. We'll clarify this in a revision.
>
> &nbsp;
>
> **Prompt selection**
>
> Our quantitative evaluation closely follows Visual Anagrams [38], randomly selecting 5 discrete (onset-based) and 5 continuous sound classes from VGGSound Common as audio prompts. For image prompts, we randomly chose 5 object and 5 scene classes, creating 100 prompt pairs through random combinations for evaluation and generating 10 samples for each. Our method scales easily for more prompts, whereas SDS baselines took more than 1,000 hours in total to generate current results for evaluation, taking two hours per example. Therefore, we kept the evaluation to a manageable scale.
>
> &nbsp;
>
> **Phrase of adversarial**
>
> We thank the reviewer's suggestion. By this phrase, we meant spectrograms that the vocoder ignores, i.e., generating waveforms that do not match the inputs. We will rephrase this in a revision.
>
> &nbsp;
>
> We hope these comments have addressed the reviewer’s questions. We would like to thank them for their time and ask that they consider raising their score for our paper.
>
> [A] Mynarski & McDermott. *Learning Mid-Level Auditory Codes from Natural Sound Statistics*, Neural Computation 2018.
>
> [B] McDermott & Simoncelli. *Sound texture perception via statistics of the auditory periphery: evidence from sound synthesis.* Neuron 2011.

---

> ### Comment · Reviewer_quDM · 2024-08-11
> **Response to rebuttal**
>
> Thanks for the reponses. Could you comment on/clarify the following?
>
> 1. "Quantative metrics": how do the baslines perform on the combined metric?
>
> 2. "Human study": is the idea of hand-picking for comparative human evaluation of generative models common? I think a better alternative would be to run the models with the same prompts multiple times and then compare all possible pairs, for computing the win/loss rates. It's undoubtedly more expensive but potentially more 'foolproof' .

---

> > ### Author Response · Authors · 2024-08-12
> >
> > We thank the reviewers for their time. Below are our clarifications:
> >
> > 1. **Quantitative metrics**: we provide the combined score (CLIP+CLAP) for baselines below:
> >
> > | Method                          |         Modality         |       CLIP + CLAP    |
> > |:---------------------------------|:------------------:|:----------:|
> > | Stable Diffusion | $\mathcal{V}$  | 1.0       |
> > | Auffusion | $\mathcal{A}$  | 1.0  |
> > | Imprint                         | $\mathcal{A}$ \& $\mathcal{V}$ | 1.04  |
> > | SDS                             | $\mathcal{A}$ \& $\mathcal{V}$ | 0.70  |
> > | Ours                            | $\mathcal{A}$ \& $\mathcal{V}$ | **1.15** |
> >
> >
> > 2. **Human study**:
> >
> > Due to constraints in the method or experiments, it is relatively common for human studies to not be completely random. We provide a short list of citations and explanations below.
> >
> > In our case, we specifically choose to study the best-case scenario for the following reasons:
> >
> > - "Images that sound" are quite hard to generate. (In fact, prior to this work, it was not clear at all that they even existed.) Not all text prompt pairs give good results, and some prompt pairs are just impossible. As such, we envisioned that a user would use our method to iteratively sample multiple times, choosing the result that they preferred most. To mimic this use case, we designed our human study to quantify the best-case results.
> >
> > - We find that it is very hard to evaluate "images that sound" on Amazon Mechanical Turk, primarily due to the fact that almost all participants did not understand the concept of a spectrogram. As a result, it is quite difficult to get precise evaluation metrics when using random results, and financially prohibitive to reduce error bars to reasonable levels. The best-case evaluation circumvents these difficulties.
> >
> > Moreover, we point out that we do in fact evaluate random results systematically and quantitatively, in a scalable fashion, with CLIP and CLAP metrics in Table 1 and FID and FAD metrics in the general rebuttal response above. We would be willing to move the human study to the appendix if the reviewer believes it would improve the manuscript.
> >
> > [1] "PhysDreamer: Physics-Based Interaction with 3D Objects via Video Generation", Zhang et al. conduct human studies on specifically 7 chosen and captured environments.
> >
> > [2] "WonderJourney: Going from Anywhere to Everywhere", Yu et al. use a total of 6 hand-designed videos for human evaluation.
> >
> > [3] "TEXTure: Text-Guided Texturing of 3D Shapes", Richardson et al. specifically pick 10 text prompts to evaluate on.
> >
> > [4] "DreamGaussian4D: Generative 4D Gaussian Splatting", Ren et al. conduct a human study on 12 specifically chosen images.
> >
> > &nbsp;
> >
> > We hope these comments have addressed the reviewer’s questions.

---

> ### Comment · Reviewer_quDM · 2024-08-14
> **Response to rebuttal 2**
>
> Thanks for the additional table. I would urge the authors to add all additional results and clarifications to the next draft.
>
> Regarding hand-picking outputs for human evaluation, I skimmed all 4 papers referred to in the latest response from the authors, but couldn't find any mention of hand-picking outputs. They do control the inputs, but that's in essence similar to controlling the text prompts in this work. However, the other arguments made by the authors are not very unreasonable. As for pushing the user study to supp., I would advise against it, as human user study is arguably the best way to evaluate these generative models, as shown in other papers in this area, including the ones cited in the paper and the rebuttal responses, as well, especially when their applications are mostly artistic in nature.

---

> > ### Author Response · Authors · 2024-08-14
> > **Author Reply**
> >
> > We thank the reviewer for their quick response. We will include all additional results and clarifications in our manuscript as the reviewer suggests. Additionally, we will keep the human study in the paper also as suggested by the reviewer, but make it abundantly clear that we are evaluating the best-case performance of all methods, and clearly explain the rational for doing so as listed above. We agree with the analysis that human studies are the best way to evaluate methods that are artistic in nature, as is the case with our method. We also agree with the reviewer that the cited papers do not explicitly mention hand-picking. We cite them to show that it is common for human evaluations to not be entirely random, in terms of prompts chosen, but also in terms of model inputs in general.

---

### Official Review · Reviewer_RECb · 2024-07-12

**Soundness:** 3
**Presentation:** 4
**Contribution:** 3
**Rating:** 7
**Confidence:** 4

**Summary:**

The authors propose to leverage pre-trained text-to-image and text-to-spectrogram diffusion models, and de-noise noisy latents with both models in parallel during reverse process. They show that the proposed method can generate spectrogram aligned with audio prompt while having visual appearance of the image prompt with quantitative analysis as well as human study.

**Strengths:**

- Proposed method has very interesting application, and can potentially provide new research direction in multimodal representation learning and generation.
- Proposed method leverage pre-trained diffusion models in both modalities to achieve listenable spectrograms with meaningful visual semantic properties, which has application in creative domain.
- This paper is clearly written and easy to follow, human study provides different perspective of proposed method compared to other baselines, and ablation study for vocoder, warm-starting and guidance scale presented in section 4.5 provide more thorough heuristics.

**Weaknesses:**

- Text prompts used for generating both spectrogram and image are mostly templating based, and composed only with simple object nouns. It would be interesting to show some generated examples on more complicated prompts with compositions of objects, this helps to provide a view into how proposed method can be scaled up for future directions.
- There is lacking in depth discussion in the choice of including "grayscale", "lithograph style", "black background", and the effect of excluding these style words. It might help to provide potential future directions in controlling the styles between the two modalities.

**Questions:**

- How are text prompts for both modalities (y_v, and y_a) selected, in Table 2, some of the pairs are the same objects, some of them are different, are there any rationale how these selections are made?

**Limitations:**

- The text prompts curated in this work is limited to only a few objects, it would be beneficial to include some methods to curate a richer and diverse prompts.

---

> ### Author Rebuttal · Authors · 2024-08-06
>
> We thank the reviewer for their positive and comprehensive feedback. Below are our responses.
>
> &nbsp;
>
> **Text prompt design**
>
> We note that our paper already contains some examples with prompts that describe visual scenes, such as in Fig. 8, where we use the relatively complex text prompt of `"a painting of a blooming garden with many birds, grayscale"`. We intentionally used simple text prompts with object or scene nouns for most experiments to allow the diffusion models to have *more flexibility* during the denoising process, since we aim to sample from the intersection of the image and spectrogram distributions. In this setting, the models are much more constrained than they would be in normal synthesis. Introducing more complex prompts could push the models to sample from even more constrained distributions where *images that sound* are less likely to exist, leading to less satisfactory results.
>
> &nbsp;
>
> **Style words**
>
> Visual diffusion models are capable of generating RGB images, but spectrograms are only one channel. We therefore use style words like `grayscale` or `black background` to nudge the image denoising process toward a distribution that matches the spectrograms. As suggested by the reviewer, we conducted ablation experiments by removing the grayscale style word. The results are shown in the table below. The model produces similar results, but (as expected) the image quality slightly decreases while the audio quality slightly improves.
>
> | Exp | CLIP ↑| CLAP ↑ | FID ↓  |  FAD ↓ |
> |:----------|:----------:|:----------:|:----------:|:----------:|
> | w/o `grayscale` | 28.1 | **33.7** |   237.12  |  **18.09** |
> | w/ `grayscale` | **28.2**  | 33.5  |  **226.46** |  19.21 |
> &nbsp;
>
> **Prompt selection**
>
> For human studies, we manually create prompts based on the prompt banks from the quantitative evaluation by augmenting them and ensuring semantic correspondence between image and audio to create artistic examples for evaluation. Please see Tab.5 in the appendix for the exact prompts used.
>
> &nbsp;
>
> We thank the reviewer for their time and hope these comments have addressed their questions.

---

> > ### Comment · Reviewer_RECb · 2024-08-12
> >
> > Thank to the authors for your answers. I still think this is an interesting idea and I am keeping my rating.

---

> > > ### Author Response · Authors · 2024-08-12
> > >
> > > We thank the reviewer for recommending acceptance. We appreciate this discussion and will incorporate the suggestions in our final version.

---

### Official Review · Reviewer_mkQc · 2024-07-12

**Soundness:** 3
**Presentation:** 3
**Contribution:** 3
**Rating:** 7
**Confidence:** 4

**Summary:**

This paper proposes a very creative idea, synthesize spectrograms that simultaneously look like natural images and also sound like natural audio, which they call images that sound. The method is rather simple, and leverages two pre-trained diffusion models, one text-to-image and the other text-to-spectrogram. A method is proposed to leverage the shared latent space and generate samples that are likely under both models. Both qualitative and quantitative results are provided, demonstrating the effectivenss of the proposed method.

**Strengths:**

- The biggest strength of the paper is the idea, very beautiful! Though making such kind of art is not completely new, this is the first method that did this as a scalable task. It nicely leverage state-of-the-art diffusion techniques and creates a model that can generate such artistic images that sound.

- The paper is also very nicely written, with clear motivation, problem statement and formulation, and nice illustrations and figures.

- The related work is complete and sufficient details are provided for reproducibility.

- The paper presents both quantiative and qualitative evaluation with nice ablation study, demonstrating the effectiveness of the proposed method.

**Weaknesses:**

There are no major weaknesses. One can argue that the method is too simple: it's basicially just combining existing techniques on diffusion, leveraging pre-trained diffusion models StableDiffusion and Auffusion, and also getting inspirations from recent image diffusion papers. Having said that, the system just works and it's more an idea paper, so I don't mind it's a simple method.

Another weakness or question is that it would be good to provide some analysis of the shared latent space. Is there way to better interpret it? For example, would it be possible to doing some interporation in the latent space such that we can see some meangingful changes in both the image domain and audio domain.

Apart from just being cool, it would also be nice to include more discussions on some more concrete potential applications of such a system.

**Questions:**

See weakness.

**Limitations:**

Yes, the authors have faithfully discussed the limitations of the proposed framework.

---

> ### Author Rebuttal · Authors · 2024-08-06
>
> We thank the reviewer for their positive and valuable feedback. Below are our responses.
>
> &nbsp;
>
> **Simplicity**
>
> We appreciate the reviewer's comment that this is a creative idea. We believe that its simplicity is in fact a major strength. While our method is simple, it is not obvious that noise estimates from diffusion models of two different modalities can be combined together during the reverse diffusion process. To our knowledge, we are the first to combine diffusion models from different modalities for multimodal composition. That this can be done by using applying ideas from compositional generation to a new domain is a benefit, rather than a drawback.
>
> &nbsp;
>
> **Shared latent space**
>
> We thank the reviewer for the suggestion. We use the same pretrained VAE encoder and decoder to map between the latent and pixel spaces for the reverse diffusion process. Since we rely on an existing latent space, we did not specifically explore its interpolation capabilities. However, we see this as an interesting direction for future work.
>
> &nbsp;
>
> **Potential applications**
>
> We thank the reviewer for the suggestion. Our work focuses on exploring the ``intersection" between the distribution of spectrograms and images, with *images that sound* being one artistic application. As we discussed in the paper, our approach could potentially be used in steganography to secretly embed images within audio for message delivery or vice versa.
>
> &nbsp;
>
> We thank the reviewer for their time and hope these comments have addressed their questions.

---

> > ### Comment · Reviewer_mkQc · 2024-08-12
> >
> > Thank the authors for the additional clarifications. I don't have further questions at this point.

---

> > > ### Author Response · Authors · 2024-08-12
> > >
> > > We thank the reviewer for recommending acceptance and valuable feedbacks for this work.

---

### Author Rebuttal · Authors · 2024-08-06

We thank the reviewers for their thorough comments and appreciate the recognition of the creativity of our work, described as "a beautiful idea" (mkQc) and "an interesting topic" (RECb, quDM, w6xJ). The acknowledgment of the "thorough evaluation" (mkQc, RECb, quDM) of our method and baselines is also valued. We answer some common questions below and provide new experimental results.

&nbsp;

**Our Contributions**

We want to emphasize that our paper goes beyond just a novel application of diffusion models. It also pioneers the exploration of the intersection between two very different distributions (images and audio spectrograms), a domain that has not been explored before. We use learned image and audio distributions from diffusion models to probe the overlapped distribution between these two modalities for free.

&nbsp;

**FID and FAD Evaluation**

Following reviewer quDM’s suggestion, we evaluated FID and FAD scores using generated examples from Stable Diffusion and Auffusion as reference sets respectively. As shown in the table below, our approach achieves the best performance. Note that FID and FAD are distribution-based metrics, and as our task focuses on generating examples that lie in a small subset of the natural image and spectrogram distribution, higher FID scores, in general, are expected.

| Method                          |         Modality         |       FID ↓    |      FAD ↓   |
|:---------------------------------|:------------------:|:----------:|:---------:|
| Stable Diffusion | $\mathcal{V}$  | --       | 41.74   |
| Auffusion | $\mathcal{A}$  | 290.29   | --      |
| Imprint                         | $\mathcal{A}$ \& $\mathcal{V}$ | 244.84   | 29.42   |
| SDS                             | $\mathcal{A}$ \& $\mathcal{V}$ | 273.03   | 32.57   |
| Ours                            | $\mathcal{A}$ \& $\mathcal{V}$ | **226.46** | **19.21** |

---

### Comment · Area_Chair_mR8v · 2024-08-12
**Author-reviewer discussion ending soon**

Dear authors and reviewers,

Thank you for the robust discussions so far. We only have about a day left in the author-reviewer discussion period, so please prioritize any remaining questions or responses.

In particular, it would be valuable to hear from reviewers mkQc and RECb. Could you please respond to acknowledge that you have read and considered the reviews and author rebuttal, and post any remaining questions that would be useful to discuss with the authors? Have the author responses addressed your concerns?

Thanks in advance.

All the best,
Your AC

---

### Decision · Program_Chairs · 2024-09-25

**Decision:**

Accept (poster)

**Comment:**

This paper describes a method to generate spectrograms called "images that sound", where the spectrogram of perceptually natural audio looks like a natural corresponding image. This suggests that the distributions of natural images and spectrograms of natural sounds have surprising overlap.

Reviewers note that the paper is very well-written and appreciate the "beautiful idea" that is the core contribution of the paper. The majority of reviewers find the objective and subjective evaluations to be thorough, complete, and convincing. Sufficient details are provided for replication.

The main weaknesses brought up by reviewers include concern about novelty, since the paper combines two prior latent diffusion models and is perhaps overlay simple; I tend to disagree that this is a major point of weakness, since this is more of an idea paper. Another point of concern is about general applicability of this method, as well as the quality of generated outputs (I tend to think quality of generated output isn't as much of a concern, given this is a challenging task, and more of an idea paper). The more critical reviewers pointed out a number of concerns including lack of clarity in some details, important of shared latents, meaningfulness of quantitative metrics, lack of standard metrics, some lack of clarity in human study, parameters of generative model sampling, and some other miscellaneous details. I think the author rebuttal and discussion provided clarity for these issues, and I encourage the authors to include these clarifications in the camera-ready draft.

Overall, with a consensus of reviewers recommending acceptance, I think this is a paper of interest to many audio, vision, and audio-visual researchers at NeurIPS that proposes and validates an interesting idea, and should thus be accepted.